# IoT Leak Detection System for Onshore Oil Pipeline Based on Thermography

**DOI:** 10.3390/s24216960

**Published:** 2024-10-30

**Authors:** Danielle Mascarenhas Maia, João Vitor Silva Mendes, João Pedro Almeida Miranda Silva, Rodrigo Freire Bastos, Matheus dos Santos Silva, Reinaldo Coelho Mirre, Thamiles Rodrigues de Melo, Herman Augusto Lepikson

**Affiliations:** Postgraduate Program in Sustainable Development (MPDS), SENAI CIMATEC University Center, Orlando Gomes Avenue, 1845, Piatã, Salvador 41650-010, BA, Brazil; joao.mendes@fbest.org.br (J.V.S.M.); joao.silva7@aln.senaicimatec.edu.br (J.P.A.M.S.); rodrigo.bastos@aln.senaicimatec.edu.br (R.F.B.); matheus91@ba.estudante.senai.br (M.d.S.S.); reinaldo.mirre@fbter.org.br (R.C.M.); thamiles.melo@fieb.org.br (T.R.d.M.); herman.lepikson@fieb.org.br (H.A.L.)

**Keywords:** pipelines, leak detection, onshore oil fields, Internet of Things

## Abstract

The vast expanses of remote onshore areas in oil-producing countries are home to a network of flow and collection pipelines that are susceptible to leaks. Most of these areas lack the infrastructure to enable the use of remote monitoring systems equipped with sensors and real-time data analysis to provide early detection of anomalies. This paper proposes a proof of concept for a monitoring system based on the Internet of Things (IoT) for real-time detection of pipeline leaks in onshore oil production fields. The proposed system, based on a thermal imaging leak detection method, informs the operator of the system’s operating status via a web page. The leak detection system communicates via a Zigbee network between the IoT devices and a 4G mobile network. The results of the tests carried out show that a visual and automatic IoT-based leak detection system is possible and plausible. The proposed leak detection system enables supervisors at remote stations and field workers to monitor the operating status of pipelines via computers, tablets, or smartphones, regardless of where they are.

## 1. Introduction

In the vast expanses of remote onshore areas where the rugged landscape meets the pristine wilderness lies a network of lifelines critical to modern civilization: oil pipelines. Approximately two-thirds of oil and gas around the world is delivered from coastal areas and transported through pipelines [1]. These arteries of industry pump precious resources across miles of terrain, fueling economies and powering nations [2,3]. Nevertheless, amid this vital infrastructure lurks a silent threat: the possibility of leaks. Pipeline leaks are the result of various causes, including natural force damage, corrosion, vandalism, and incorrect operation conditions [4,5,6].

Fortunately, advancements in technology are being actively conducted in the oil and gas industry for protection against pipeline leaks. Remote monitoring systems, equipped with sensors and real-time data analysis, provide early detection of anomalies, allowing operators to swiftly identify and mitigate potential leaks before they escalate into full-blown disasters [7,8].

However, in countries like Brazil, for example, where most of the onshore oil production occurs in remote regions of the country, access and communications are limited [9]. Most of these areas lack infrastructure, such as roads, electricity grids, and telephone networks [10], which makes it difficult to monitor the flow of oil production.

It is not uncommon, even where there is a local network, for there to be a weak signal, slow browsing speeds, and frequent connection drops. Communicating information in this environment is a challenge when considering how important an effective monitoring system is for mitigating pipeline leakage problems.

Monitoring pipelines and detecting leaks are important activities for oil production operators, whether due to the increased stringency of environmental legislation or the reliability of the system. The earlier the detections, the quicker the responses, allowing for more effective intervention in the system [11].

Currently, the monitoring of many pipelines in the oil production system of onshore fields is carried out by means of a visual inspection, in which an operator must go to certain sections of the operation line and observe the areas that catch his eye. Visual inspections can also be carried out by third parties hired just for this purpose, or an analysis of secondary data from the supervisory system implemented, such as SCADA (Supervisory Control and Data Acquisition), of suspicious stretches of pipelines.

New initiatives on fast and accurate monitoring and/or detection of leaks are essential so that the oil industry can not only debate the issue but remain competitive in a sustainable way. In addition to pipelines being in large remote areas, they are interconnected with old systems that operate without human assistance continuously present.

In this context, the Internet of Things (IoT) is currently an effective possibility for monitoring the intended environment. The aim is to ensure that physical devices with embedded electronics, software, sensors, actuators, and network connectivity can identify, collect, and exchange data with each other [12]. Regardless of its functionality, this entire system (processor, a communication transceiver, and a power supply) must be appropriately designed to be inserted in each environment.

Nevertheless, intelligent and reliable monitoring is highly desirable, and IoT is the key to establishing such monitoring through low-cost micro-controlled devices [12], which can be wirelessly networked, making onshore oil field applications feasible.

Thus, taking preventive action will allow the sector to contribute more and more to the national industry and essential partnerships with governments in contributing to the challenge of achieving the Sustainable Development Goals (SDGs) established by the United Nations for the 2030 agenda.

Considering this scenario, contributing to the development of new pipeline monitoring systems in onshore fields for leak detection can play a key role in directly promoting various SDGs, such as SDG 3 (Good Health and Well-being), SDG 6 (Clean Water and Sanitation), SDG 7 (Affordable Clean Energy), SDG 12 (Responsible Consumption and Production) and SDG 15 (Life on Land). In addition, the monitoring of pipelines in onshore fields is significant for the sustainable growth of the sector over the years.

This work proposes an IoT-based monitoring system to assist in real-time leak detection in pipelines to contribute to the area of oil production in onshore fields. The proposed architecture for the IoT monitoring system is shown in Section 2. The results obtained from experimental tests in the lab are described in Section 3, as well as the discussion and conclusion reported in Section 4 and Section 5, respectively.

## 2. Methods

### 2.1. Monitoring Systems for Leak Detection

The implementation of an IoT monitoring system in remote onshore oil production fields for leak detection basically requires the definition of three pillars: the choice of method and devices, a communication and connectivity network, and a control system that guarantees coverage in remote and hard-to-reach areas.

In the quest to pinpoint the acceptable leak detection strategy for delineating the specifications of a remote leak monitoring system aimed at onshore oil production fields, a methodological approach was carried out comprising six fundamental steps [13]. These steps initially involve defining the magnitude of the problem, followed by defining the objectives that guide the proposed solution.

Subsequently, the focus shifts to the characterization of the object to be monitored, requiring an understanding of its main attributes to facilitate the selection of appropriate detection methods. Outlining the requirements of the detection system appears to be a crucial step, serving as a basis for decision-making. The selection process culminates in a comprehensive analysis of available detection methods, their underlying principles, advantages, and disadvantages, paving the way for the careful selection of the most compatible method and sensors to meet the system requirements [13].

It can be observed that although the oil and gas industry uses a wide range of techniques to detect leaks, the application of Internet of Things (IoT)-based detection and collection technology that offers uninterrupted, real-time monitoring along the line route is rare or non-existent. In this case, the objective is to develop an architecture for a monitoring system based on the Internet of Things (IoT) to detect leaks in flowlines and gathering pipelines.

A leak detection method can be determined to provide a reliable and applicable mechanism by defining the desirable requirements for the monitoring system. The proposed method should ensure safe and remote leak inspection and offer an automatic and autonomous inspection mechanism. The system must be robust against environmental noise and capable of detecting multiple simultaneous leaks, as well as pinpointing the exact location of each leak. Additionally, it should rely minimally on the physical properties of the transported fluid and the material of the pipes, making it applicable regardless of the fluid’s physical state and the pipe materials. Finally, it must be considered that production and collection lines transport multiphase fluids, including crude oil, water, and gases.

According to the main related works in this thematic, there are some classifications of pipeline leak detection methods due to the utilization of diverse physical principles, approaches, and applications [11]. The methods can be classified according to the degree of automation (automatic, semi-automatic, and manual detection techniques) or intuitive degree of detection data (direct and indirect detection) [14]. Other approaches divide them into three categories: hardware-based methods, software-based methods, and biological methods [15], or into internal systems, external systems, and visual systems [11,16].

The most common category is that which classifies hardware- and software-based methods [14]. The use of devices designed by physical principles in pipelines refers to hardware-based detection methods. They are generally sensitive to small leaks and precise in their location. The instrumentation is usually run along the entire length of the pipeline, which helps in the detection of large and small leaks in a timely manner and allows the detection of a leak anywhere along the pipeline [14].

Table 1 lists the hardware-based methods that can be applied to both oil and gas pipelines.

The pipes used in onshore production fields are long-distance oil and gas pipelines, and among the detection techniques in Table 1, only the following can be chosen: Acoustic Emission, Ultrasonic, Guided Wave Ultrasonic, Distributed Optical Fiber, SmartBall, Vapor Sampling, and Thermal Imaging. Table 2 describes each method with its advantages and disadvantages.

In the process of analyzing the advantages and limitations of each hardware-based detection method and in accordance with desirable requirements, the thermal imaging method was selected through the application of thermal cameras as it facilitates the automatic inspection and detection of leaks, as it is a robust system to environmental noise and be able to detect small leaks.

The thermal imaging leak detection method provides a reliable and applicable detection mechanism and does not require the collection of information on the physical properties of the fluid, pipeline material, or process parameters [24]. As opposed to the acoustic and ultrasonic methods, which require prior knowledge of the physical characteristics of the pipe, the wave propagation, and its operating modes in relation to each cause generating the leak [19]. Meanwhile, thermal cameras have been used as a remote inspection method in various industrial applications and have been evaluated as facilitators in the automatic inspection of leaks, constituting a robust system capable of detecting even small leaks [13,24].

The proposed design is based on IoT devices that have autonomous hardware, reduced dimensions, and maintenance at spaced intervals. In this way, they form a wireless sensor network (WSN) ideal for collecting data from the environment and automating tasks. The use of WSN in oil and gas pipeline monitoring cases has a topology based on a high density of sensors, which are distributed along the pipelines and organized into groups of nodes to control certain segments of the pipeline. Adopting WSN technologies in pipeline monitoring is an interesting option for preventing the danger caused by leaks [25].

### 2.2. Proposed Architecture

For monitoring oil pipelines, an IoT architectural solution for a thermography-based leak detection system is proposed in Figure 1.

The proposed architecture is composed of three layers, namely the perception, network, and application layers. The perception layer is the physical layer, which has sensors for sensing and gathering information about the environment. The network layer is responsible for connecting to other smart things, network devices, and servers. The application layer is responsible for delivering application-specific services to the user [26].

In principle, thermal cameras are devices that make up the perception layer of the IoT system, with processing, storage, and communication capabilities. Thermal cameras produce images from the infrared radiation of the object at any angle and area. These cameras distinguish the level of the image using different colors that give an idea of the object and its shape, based on the fundamental principle of thermodynamics that energy flows from the warm area to the cooler area [22]. The function of these devices is to photograph the pipelines at each time interval. To share the data collected, a wireless network architecture must be compatible with the specifics of the environment and the type of data packet to be transferred.

The network layer of the IoT system is responsible for enabling the exchange of data between cameras and the IoT platform by sending and routing packets. Most wireless network systems use radio frequency modulation. Licensing for the use of radio bands is regulated by government authorities. In addition to being able to connect to the internet, the application layer of the IoT system has specific requirements, such as low data transfer rates, low power consumption, and cost-effectiveness [27].

Therefore, the system starts by automatically triggering a thermal camera to photograph the flow and gathering lines in the production fields. This captured image is sent to a gateway via a wireless communication network. It is at this stage that the image is processed. Regardless of the processing information, if there is or not a leak, it will be stored on an IoT platform via the Internet, consequently allowing operators at remote stations to monitor the operational status of the pipelines via computers, tablets, or smartphones. In the event of a leak in the system, the user will also receive a georeferenced image of the site, indicating the location of the leak and its visual status.

### 2.3. Proof of Concept (PoC)

An PoC was developed to validate the proposed IoT monitoring system for leak detection. Figure 2a shows the IoT monitoring system and the hydraulic system, which was used to simulate a pipe leak. The operation of the PoC begins when the centrifugal pump is activated after the electric motor is started, as shown in the P&ID diagram (Figure 2b). The pump then begins to suck water from the reservoir by accelerating the impeller. The kinetic energy transmitted by the impeller is converted into pressure energy, propelling the water through the pipe network in an open circuit, manually controlled. As the water circulates, it passes through valves and pressure gauges, barely connected to simulate leakage by dripping.

The IoT monitoring system is powered by an individual power supply. When the system is switched on, the Flir ADK Thermal Vision camera automatically photographs the pipe under study. This camera offers a cost-effective approach to the development of automotive thermal vision. The ADK’s rugged, IP67-rated enclosure incorporates a heated window for all-weather driving, and the Boson™ thermal sensor is a fraction of the size of current night vision systems. With USB, GMSL, Ethernet, and FPD-Link interfaces, installation is plug-and-play easy. The camera is configured and controlled using a Raspberry Pi 3 Model B microcomputer, connected via USB. This captured image is sent to the gateway (a second microcomputer) via the communication established between the Xbee Zigbee S2 modules. The diagram in Figure 3 shows the technologies used in each layer of the IoT system developed.

The layout of IoT devices is based on mesh topology with interconnection via the ZigBee protocol. This approach allows each device to receive and relay data even in remote locations. ZigBee is specially designed for domestic and industrial applications. It is an open global standard for implementing low-cost, low-data-rate, short-range wireless networks with a long battery life, enabling them to be used in monitoring and control implementations. In addition, due to the easy accessibility of the modules, this technology was interesting for enabling data communication in the proposed system.

As the ZigBee network is intended for wireless sensor network communication, a means of communication must also be defined for transporting the data over the Internet to increase the system’s flexibility. One alternative is to use the mobile network technologies available close to the production area, such as 3G or 4G or even new generations of 5G or 6G if already available. The advantage of these mobile networks is that they take advantage of existing infrastructure.

To send the captured image, it is necessary to convert the JPG image into a grayscale matrix, due to communication restrictions. Then, this matrix is converted into a one-dimensional vector, which is sent to the communication receiver module through the microcomputer’s serial port. Each packet contains a part of the data to be sent. Packets are sent sequentially until the frame is completed. The average time to avoid losing information between each packet sending is two seconds.

After the image has been sent and received by the second microcomputer, it needs to be reconstructed so that it can be viewed by the operator after receiving it over the Internet. This layer also performs some image processing steps to automatically obtain information if there is or is not a leak in the system.

At this stage, the first step is to apply a color mapping to the gray scale image. The choice of color map can vary from programmer to programmer, as can the types of systems being analyzed. After coloring the image, a mask is applied for future selection of the region of interest in the captured image based on the color that will represent the area under study.

The mask works as a filter, in which it filters out everything outside the analyzed area and maintains the frequency of the color chosen to identify the region of interest. Recognizing the area by color makes it easier to apply the similarity segmentation technique.

One of the main similarity approaches is based on thresholding. Mathematically, the technique can be described as an image processing technique in which an input image *f*(*x*, *y*) of N gray levels produces an output image *g*(*x*, *y*), called the thresholded image, whose number of gray levels is less than N. Normally, *g*(*x*, *y*) has two gray levels [28]:$$g\left(x,y\right)=\left\{\begin{array}{cc} 1 & \mathrm{if}\;f(x)\geq \;T\\ 0 & \mathrm{if}\;f(x)<T \end{array}\right.$$
where pixels labeled 1 correspond to objects and 0 to the background; *T* is a predefined gray tone value called the threshold. This technique therefore allows the image to be binarized.

Furthermore, a binary image is an image made up of pixels that can be exactly one of two colors. This means that each pixel is stored as a single bit 0 or 1. Binary images can be stored in memory as bitmaps [29].

To identify both the pipe and the appearance of the leak in the image, the contour tracing technique is used. This technique will allow you to find a white object, in this case a pipe or leak, on a black background. Next, a filter is run to identify the number of contours found in the image defined from a minimum and maximum pixel area. These areas with contours are listed and sorted. This logic will be applied twice: the first to locate the pipe in the image and the second to identify leaks near the pipe.

In the first stage, to find the pipe, it is assumed that the image area is larger than the pipe area. The interest is precisely on the pipe and its surrounding regions, so selecting the smallest area will mean the pipe. Identifying smaller areas, such as the pipe, prevents other objects in the image from interfering with the selection. With this new image, limited by the size of the identified pipe, the smaller area contour is selected again, now signifying the leak.

To ensure that the area identified is in fact a fluid, the Canny edge detection technique is applied, which allows the noise to be smoothed out and the edges of the contours to be located. This requires repeating the mask and contour application stage so that the areas corresponding to the leak can be detected and selected.

Next, using the Secure Socket Shell (SSH) network protocol, a secure connection is established with an online repository called PIPECOM (https://github.com/theussant/PIPECOM.git, accessed on 4 September 2024 )created on the GitHub platform, which stores the data received and manages the system’s algorithms. The IoT platform layer developed for the leak detection and monitoring architecture is finalized and integrated with a web page so that the end user can view the data in a more user-friendly way. The page was implemented using the Vercel platform, allowing online access to the system via a web link (https://pipecom.vercel.app, accessed on 4 September 2024), as well as monitoring its status.

Finally, by accessing the developed website, information is obtained on if there is or is not a leak in the pipe, as well as receiving the captured image for visual validation of the message.

### 2.4. Validation and Testing

The final tests to verify the integrity of the system, through the correct flow of data between the layers, took place after the development and configurations of all the layers of the IoT system and their respective integrations had been completed.

At this stage, it is necessary to determine the limit distance that the camera can be in relation to the hydraulic system. This maximum distance will be where the IoT device will be able to identify the pipe and detect some kind of leak; in addition, the transmission time of the images via the Zigbee communication modules for the chosen distance can be observed.

In this scenario, the camera was positioned at different points throughout the warehouse, where the hydraulic system is located. The distances of 10 m (Figure 4a), 25 m (Figure 4b), and 40 m (Figure 4c) were chosen, as these are the minimum, average, and maximum distances from the warehouse.

## 3. Results

After running the tests, the position of the camera 10 m from the hydraulic system was the maximum possible distance to see the pipe clearly in the image, as shown in Figure 5. In this case, this position detects that the system was indeed leaking, considering the pipes were only 19 mm (3/4′) in diameter.

Figure 6 shows the result of applying image processing techniques to the image captured 10 m from the hydraulic system. On the left of the image, the blue marking shows where the leak is. The other points of water droplets are splashes from the centrifugal pump’s priming funnel, which is used to ensure that the system works properly.

Selecting just one leak for a site with several leaks already allows the information to be sent to GitHub and consequently updated on the platform. Afterwards, the captured image from the system with the information that there is a leak is available on the system monitoring website, as seen in Figure 7.

The time taken to send and receive the image from the IoT device to the Gateway was approximately 40 min. No explicit image compression techniques were used in this PoC. It should be noted that the application of image compression techniques or other image transmission mechanisms in Zigbee-based sensor networks could probably be a good solution for reducing this transmission time. However, the simple image transmission logic adopted guaranteed reliable and real-time transmission considering the purpose of the application, with no loss of packets sent.

## 4. Discussion

Implementing a suitable IoT monitoring system in remote onshore oil production fields requires telecommunications solutions that guarantee coverage in remote and hard-to-reach areas, as well as providing secure and reliable communications. Determining a leak detection method to provide a reliable and applicable leak detection mechanism was made possible by defining the desirable requirements for the monitoring system.

The proposed system monitors through IoT devices with infrared cameras associated with systems with processing, storage, and communication capabilities in an IoT environment. The proposed solution also has the advantage of not requiring the collection of information on the physical properties of the fluid, pipeline material, or process parameters.

There are various models and technologies of cameras on the market. When choosing an infrared camera, there are several factors that are important to consider. First, the intended application of the equipment should be considered because, for example, the requirements of a camera intended for leak detection will necessarily be different from a camera that will be used in vehicles, which is the case of the camera used in PoC. The FLIR ADK™ model was developed for advanced driver assistance systems and autonomous vehicles for the purpose of pedestrian detection.

Despite being a camera designed to be an economical way to develop the next generation of automotive thermal vision, it is robust with an IP67 rating, has a good design for driving in any climate, and has GMSL and USB interfaces with easy installation via the plug-and-play system. However, using a camera designed specifically for leak detection would provide even better results than the one used in the tests, even if it fulfilled the application requirements of the concept.

The architecture and technologies applied demonstrate a positive impact on leak detection in remote fields, as the installation was in a shed, where the facility’s mobile telecommunications network and connection system could not reach the site. The use of the Zigbee communication network for communication between devices arranged in a mesh makes it possible to create wide coverage even in remote locations through Zigbee nodes installed throughout the production field, allowing as many devices as desired to be added to the network at a low cost and with low energy consumption.

The results obtained through the PoC developed showed that an IoT-based monitoring system for automatic visual leak detection is possible and plausible. This research and testing were based on basic image processing techniques and did not use any artificial intelligence. However, the architecture developed has the potential to work with new complementary systems running in the cloud, such as building a database that provides data on leaks for training and validating models, for example. This would have a positive impact on problem prevention and the availability of historical data for predictive management of production systems in the oil and gas sector.

Or even its application in integration with existing sensors and actuators in pipelines for more complex analyses of causes and effects that contribute to mitigating leakage problems. This demonstrates the system’s positive impact on leak prevention in the oil and gas sector and its ability to provide a track record for the predictive management of production systems regarding leaks.

## 5. Conclusions

In this work, the PoC confirmed that the proposed system offers benefits such as real-time detection, continuous monitoring, and data analysis for more assertive decision-making with a relatively low-cost solution, especially when considering the challenge of leak detection in remote onshore areas as the problem to be solved.

The proposed solution is based on devices already available on the market. The camera, the main cost item, is already used in industrial environments, which attests to its robustness. The mesh network also has multiple applications for effective communication in remote environments. The main contribution of this work was to integrate these systems through computationally intelligent algorithms that made their application effective.

It was also demonstrated that an IoT monitoring system for oil spill detection can have a significant impact on reducing environmental impact and the costs involved when compared with the means currently in use, which are expensive and not very effective. Including when considering the increasingly representative fines involved.

As a proposal for future correlated work, the application of the proposed system in real operating conditions in onshore oil production fields for a quantification of the solution, as well as the verification of the influence of the surface conditions of the test area, solar radiation, cloud cover, and ambient temperature.

Although a control system has not been proposed in this PoC, the implementation of the suggested system architecture can help to identify leaks remotely and take measures to contain them, minimizing the negative impact on the environment and the costs involved, including fines, which are increasingly representative. Thus, the proposed IoT LDS makes it possible for supervisors at remote stations and field workers to monitor the status of pipeline operations using other devices, such as computers, tablets, or smartphones, regardless of where they are.

Finally, this monitoring system development can play a key role in promoting social, economic, and environmental impacts. In addition to the direct benefits for SDGs 6, 7, 12, and 15 mentioned, it also contributes indirectly to other sustainable development goals, including 3 (Good Health and Well-Being), 9 (Industry, Innovation and Infrastructure) and 17 (Partnerships for the Goals). Reducing leaks and losses can reduce industrial accidents and the risks to human health through safety in the operation of onshore oil production lines and for the professionals involved. Therefore, by adopting more sustainable practices, companies can improve their reputation and corporate image.

## Figures and Tables

**Figure 1 sensors-24-06960-f001:**
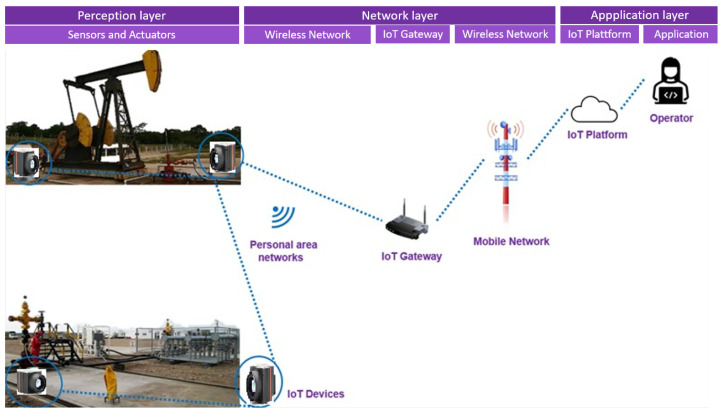
Architectural proposal.

**Figure 2 sensors-24-06960-f002:**
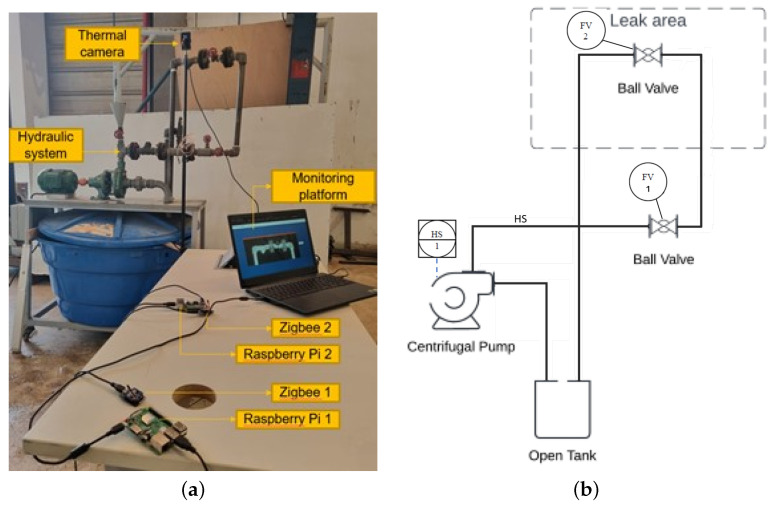
System operation. (**a**) PoC. (**b**) P&ID: hydraulic system.

**Figure 3 sensors-24-06960-f003:**
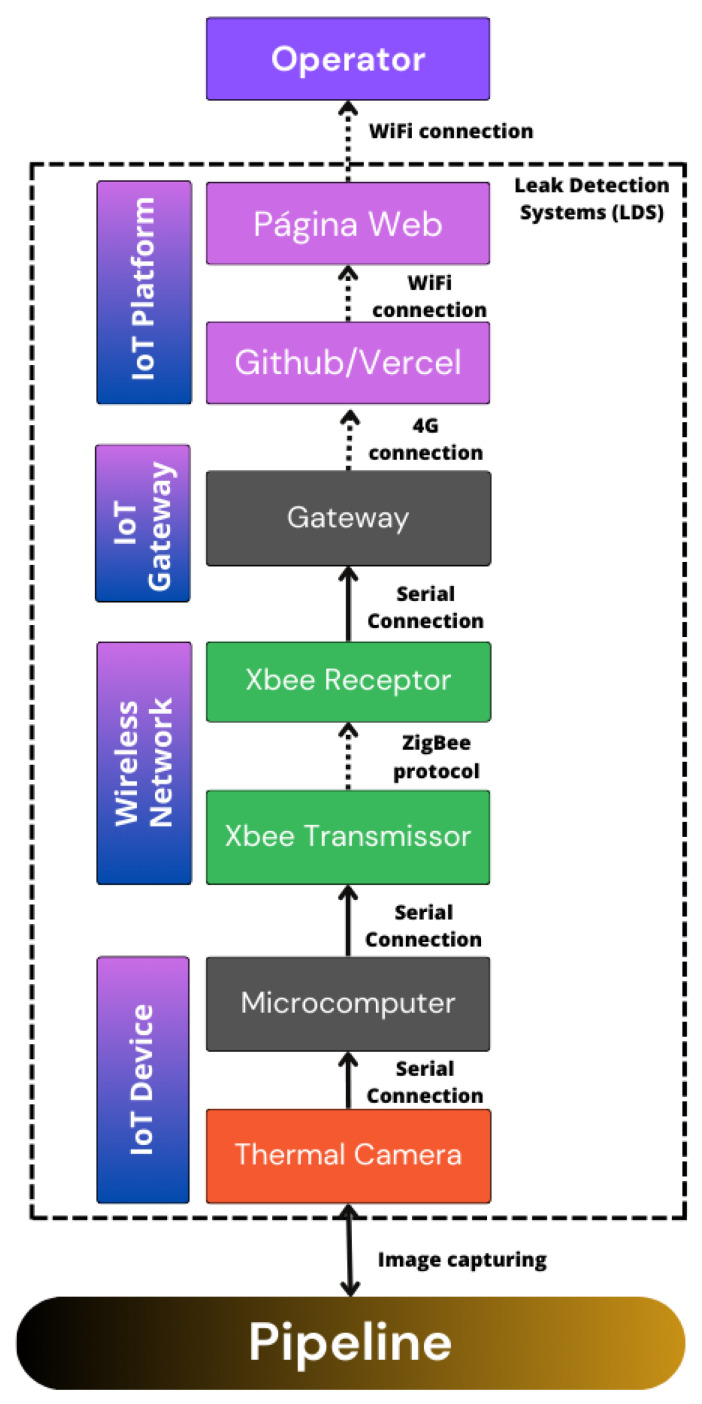
Device connection diagram.

**Figure 4 sensors-24-06960-f004:**
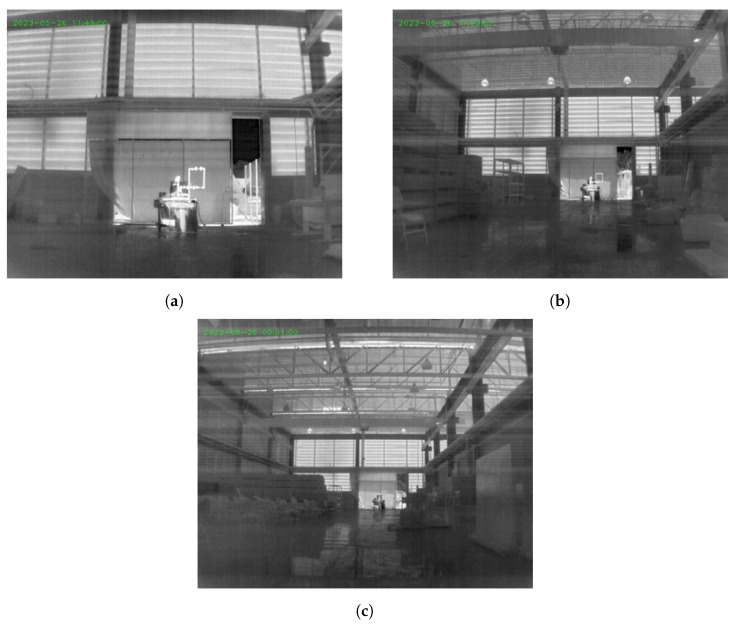
Camera positions in front of the hydraulic system. (**a**) 10 m; (**b**) 25 m; (**c**) 40 m.

**Figure 5 sensors-24-06960-f005:**
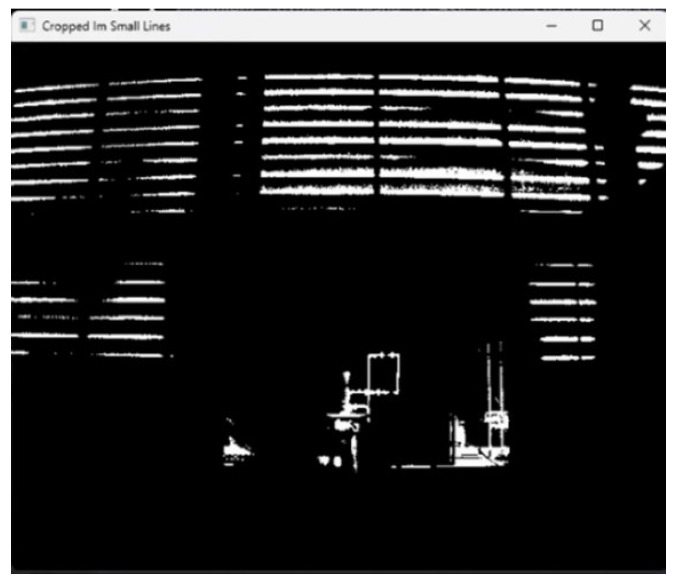
Thermal camera application and image processing.

**Figure 6 sensors-24-06960-f006:**
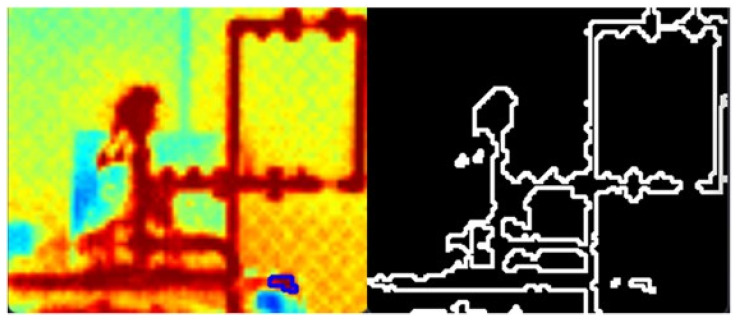
Application of the edge detection technique.

**Figure 7 sensors-24-06960-f007:**
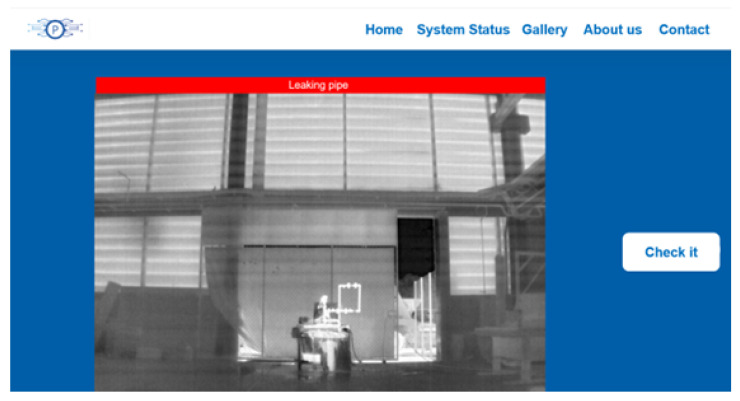
Application of the edge detection technique.

**Table 1 sensors-24-06960-t001:** Application of hardware-based methods for leak detection.

Classification	Detection Technique	GU	GL	GS	OL	OS
Hardware-based method	Lidar system	x x				
	Diode laser absorption	x	x			
	Millimeter wave radar system	x	x			
	Thermal imaging	x	x		x	
	Spectral imaging	x	x			
	Sound wave method	x	x	x	x	x
	Ultrasonic guided wave method	x	x	x	x	x
	Sonar system method			x		x
	Acoustic guided wave method	x	x	x	x	x
	Ultrasonic method	x	x	x	x	x
	Distributed optical fiber sensor method	x	x	x	x	x
	Tracer method	x	x	x		
	SmartBall method	x	x	x	x	x
	Ultrasonic flowmeter method	x	x	x	x	x
	Dynamic pressure transmitter method	x	x	x	x	x
	GPR	x	x		x	
	Cable method				x	
	Sniffer method	x	x			
	Vapor sampling method	x	x		x	

**Abbreviations:** GU—urban gas pipeline; GL—long distance gas pipeline; GS—subsea gas pipeline; OL—long distance oil pipeline; OS—subsea oil pipeline. Source: adapted from [14].

**Table 2 sensors-24-06960-t002:** Overview of the sensing methods for oil pipeline leak detection.

Methods	Description	Advantages	Disadvantages
Acoustic emission	When a leak occurs, the acoustic signal and the background noise reach the sensors and are compared with the profiles of acoustic wave signals present in the system [17].	The detection has higher sensitivity and accurate positioning [4]. It is easy to use and install as it does not require system shutdown for installation or calibration [11].	The effect of noise from other sources can easily mask the actual leak sound [11]. In addition, the type of piping material interferes with leak recognition.
Ultrasonic	The ultrasonic method is characterized by being hardware-based, acoustic, and non-invasive. A common way of applying the technique is to use a pulsar, a transducer, and a device to display the captured signals [18].	This method has a fast response and high sensitivity, low detection cost, and low power consumption [14].	Its implementation requires extensive knowledge and careful application by experienced technicians. Couplants are required to provide effective transfer of energy between the transducers and the piping. Reference standards are also needed, both for equipment calibration and for fault characterization [18].
Guided Wave Ultrasonic	For leak detection, a ring of piezoelectric transducers is installed around the duct. When there is a need to carry out the inspection, a device is attached that controls the transducers, causing them to emit pulses and receive them back [19].	This method has a shorter inspection time and less labor demand. It presents itself as a great proposal to monitor long lengths of pipeline due to the ability of guided waves (lamb waves) to propagate in materials such as steel [14].	Currently there is little variety in the market for standard devices, which have high costs. Another difficulty is related to data processing: defining the ideal frequency for the emission of guided waves and comparing it with the received signals is a process that requires specialized technical knowledge. It requires pre-calibration.
Distributed Optical Fiber	The most common choice of fiber optic sensors in hydraulic applications is scattering-based (Rayleigh and Brillouin scattering-based sensors) and wavelength-modulated (fiber Bragg grating) [20].	The sensors require no electrical power and have low optical transmission losses and little signal decay. The same fiber can be used to detect and transmit information [20].	Interference is required in the pipes to install the sensors.
SmartBall	This is a comprehensive sensing technology that includes an array of acoustic sensors, accelerometers, magnetometers, ultrasonic transmitters, temperature sensors, and more. These sensors roll inside the pipe following the fluid flow [14].	The technology can be used for pipes of any material, and the sensitivity of leakage detection is high [14].	Under different working pressures, the minimum detectable leakage of the SmartBall is different. The presence of an operator is required to insert the SmartBalls, in addition to the limitations of internal access to pipes with junctions, diversions, and manifolds, characteristic of oil production fields.
Vapor sampling	It consists of taking hydrocarbon vapor samples near the monitored pipeline to determine the oil leak based on the measured gas concentration. This method can use a vapor monitoring system or mobile detector device [14,21].	This technique is particularly suitable for detecting small concentrations of diffuse gas [14].	The cost is high and requires many sensors in long pipelines [21]. Gases such as biogas in the soil may cause false alarms [14].
Thermal Imaging	The method uses an infrared camera to analyze changes in thermal radiation around the pipeline [14] in the infrared range of 900–1400 nanometers [22].	Increasing use of technology for remote monitoring of unmanned plants market trend for the search for visual solutions. The technique is not affected by the materials or sizes of the tubes and their connections [23].	Area surface conditions, solar radiation, cloud cover, and ambient temperature can impact the capabilities of this method [23].

Source: From [13].

## Data Availability

The original data (data sheet of the devices) presented in the study are openly available in the PIPECOM (https://github.com/theussant/PIPECOM.git, accessed on 4 September 2024) repository.

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
