# Peer review of "IoT Leak Detection System for Onshore Oil Pipeline Based on Thermography"

_sensors, 2024, doi:10.3390/s24216960_

Round 1

Reviewer 1 Report

Comments and Suggestions for Authors

this paper discusses pipeline detection systems using thermal cameras however, I have the following comments

1- The paper presentation is too weak I recommend to add related work section that compares different leakage detection methods and the merit of your method   (https://www.sciencedirect.com/science/article/pii/S2667143322000464)

2- the number of references is too small 

- https://www.mdpi.com/2076-3417/13/5/2845

- https://www.sciencedirect.com/science/article/abs/pii/S0263224119300363?via%3Dihub

- https://www.sciencedirect.com/science/article/pii/S2667143323000215

-https://link.springer.com/article/10.1007/s12666-020-02002-x

- https://www.sciencedirect.com/science/article/pii/S1738573316301504

-https://ieeexplore.ieee.org/document/10592530

-https://ieeexplore.ieee.org/document/10563208

3- A comparison with previous methods should be conducted 

Comments on the Quality of English Language

English needs to be carefully revised

Author Response

Comments 1:  The paper presentation is too weak I recommend to add related work section that compares different leakage detection methods and the merit of your method.

Response 1: Thank you for pointing this out. The types of detection methods have been presented between lines 105 and 129, and they have been compared and summarized in tables 1 and 2.

Comments 2: the number of references is too small 

Response 2: Thank you for the suggestions about references, we have used some of them, as well as adding others already used in the development of the project. They were included in the introduction and methods sessions (chapter 2.1). The number of references has increased from 13 to 29, as listed in the references section. 

Comments 3: A comparison with previous methods should be conducted. 

Response 3: Agree. We made comparisons using tables 1 and 2.

Reviewer 2 Report

Comments and Suggestions for Authors

1. Improve the introduction

2. In Figure 1 the image of the supposed camera is not that of a camera.

3. In chapter 2.2, make known the software that this camera uses, the type of camera that is used and make known the characteristics of the camera.

4. Put the type of platform that is used to store the data in the cloud and make known in general terms how you can download this data.

5. Important! In chapter 2.3. I would like to see a control panel and a power panel, where the connections between the different instruments and the actuators can be adequately visualized. Even put an electrical diagram of their connections. Likewise, put the connections of the camera with the raspberry. Put the type of protocols that the camera works with and how it communicates with the raspberry. Put the code that is required to at least communicate the camera with the raspberry.

6. In chapter 2.3, include a Piping and Instrumentation Diagram (P&ID) of the process.

7. Place the GitHub link where this information comes from at the beginning of chapter 3, or simply place it in the article.

8. Improve the conclusions

9. The number of references is very small, increase it to at least 20 references.

Comments on the Quality of English Language

No comments

Author Response

Comments 1:  Improve the introduction

Response 1: New information and references have been added relating to the causes of leaks and searches for new technologies. These improvements are found from line 16 to 26 and 58 to 67.

Comments 2: In Figure 1 the image of the supposed camera is not that of a camera.

Response 2: Thank you for pointing this out. Although the image is for illustrative purposes only, given the possibility of implementing new models, we agree to modify it.

Comments 3: In chapter 2.2, make known the software that this camera uses, the type of camera that is used and make known the characteristics of the camera.

Response 3: In chapter 2 we intended to emphasize only the architecture of the system, not limiting ourselves to possible existing technologies, which is why we didn't add the characteristics of the camera in this section. We concluded that the best session to describe it was in chapter 2.3. Even the beginning of the session, from line 187 to 197, has been modified.  The data sheet of the devices is openly available in PIPECOM repository.

Comments 4: Put the type of platform that is used to store the data in the cloud and make known in general terms how you can download this data.

Response 4: Data access and algorithm management is detailed in paragraphs 380-381.

Comments 5: Important! In chapter 2.3. I would like to see a control panel and a power panel, where the connections between the different instruments and the actuators can be adequately visualized. Even put an electrical diagram of their connections. Likewise, put the connections of the camera with the raspberry. Put the type of protocols that the camera works with and how it communicates with the raspberry. Put the code that is required to at least communicate the camera with the raspberry.

Response 5: Thank you for pointing this out. The camera's communication ports and protocols with the raspberry and modules are shown in Figure 3. Nevertheless, we do not have access to the control panel and electrical panel of the hydraulic system, as we are using a platform for teaching purposes at the University.

Comments 6. In chapter 2.3, include a Piping and Instrumentation Diagram (P&ID) of the process.

Response 6: We implemented the suggestion by adding the PID diagram (Figure 2b) and adding a paragraph describing how the hydraulic system works (Line 178 to 186).

Comments 7. Place the GitHub link where this information comes from at the beginning of chapter 3, or simply place it in the article.

Response 7: Links added in Chapter 3 (Lines 255 and 264) and in the ''Data Availability Statement'' section (Line 380).

Comments 8: Improve the conclusions

Response 8:  The conclusions were improved in Chapter 5, including more information about the proposed solution contribution (Lines 341 to 373).

Comments 9. The number of references is very small, increase it to at least 20 references.

Response 9: Thank you for pointing this out. The new references were included in the introduction and methods sessions (chapter 2.1). The number of references has increased from 13 to 29, as listed in the references section. 

Round 2

Reviewer 1 Report

Comments and Suggestions for Authors

the authors reflected my comments correctly

Comments on the Quality of English Language

english is ok

Author Response

Comments 1:  the authors reflected my comments correctly

Response 1: Thank you for your feedback. We're glad to hear it, your comments and suggestions were essential to improving the quality of our work.

Reviewer 2 Report

Comments and Suggestions for Authors

1. I think they uploaded the same document as the previous one, the only thing that changes is that the supposed changes are underlined in red but there is no difference in what was requested.

2. The changes that I requested do not appear.

Please check if the version they uploaded is the same as the previous one. If it is the same, upload the version with the requested changes.

If not, please make the requested changes.

Comments on the Quality of English Language

No comments

Author Response

Comments 1:  I think they uploaded the same document as the previous one, the only thing that changes is that the supposed changes are underlined in red but there is no difference in what was requested.

Response 1: Thank you for your feedback. We apologize for any confusion caused. We did submit a revised manuscript, but it seems there was an issue with accessing the updated document. To ensure clarity, we have attached the revised manuscript again, which includes all the changes requested in the first round of review.

Comments 2:  The changes that I requested do not appear. Please check if the version they uploaded is the same as the previous one. If it is the same, upload the version with the requested changes. If not, please make the requested changes.

Response 2: To resolve this, we have attached the revised manuscript again. Additionally, we kindly ask you to consider our responses provided in the first round of review. We appreciate your understanding and look forward to your further feedback.
